# Microfinance Facility for Rural Women Entrepreneurs in Pakistan: An Empirical Analysis

**Touseef Ahmed Khan [1], Fahem Ahmed Khan [2], Qristin Violinda [3], Ilyas Aasir [1] and Sun Jian [1,\*]**

[1]  College of Economics and Management, Huazhong Agricultural University, No. 1, Shizishan Street, Hongshan District, Wuhan 430070, China; teekay.khan@yahoo.com (T.A.K.); aasirilyas@yahoo.com (I.A.)

[2]  College of Animal science, Huazhong Agricultural University, No. 1, Shizishan Street, Hongshan District, Wuhan 430070, China; faheemgenetics@yahoo.com

[3]  Faculty of Economic & Business, Universitas PGRI Semarang, Jawa Tengah 50232, Indonesia; qristinviolinda@ymail.com

\*  Correspondence: sunjianhn@mail.hzau.edu.cn

**Abstract:** Since 1990, microfinance has gained universal recognition as an essential and useful tool to address the economically productive poor and provide them with a way to come out of the vicious circle of poverty, by delivering loans and credit on flexible terms in contrast to commercial banks. Many studies from different parts of the world have shown significant economic and social uplift of recipients (both men and women) of microfinance programs. However, in recent years, some studies argue against the positive influence of microfinance and stresses that microfinance is commercialized, and it has become more of a profit generation activity than uplifting of the economically productive poor, which is one of many core objective of microfinance organizations. Many empirical studies have been done to know the effects of microfinance on the welfare of households. They are well documented, e.g., in Bangladesh and India, but only a few studies assess the microfinance effect on rural female entrepreneurs of Pakistan. The present study was carried out to empirically analyze the outcome of microfinance on Pakistan's female entrepreneurs. Women's empowerment is gauged using income and consumption as welfare indicators. The difference in difference method is applied to investigate the effects of microfinance on its recipients, which is considered a useful tool to tackle the selection bias problem. Our study result shows that microfinance programs that target women not just only increase income and consumption of female borrowers, making them financially stable, but in fact, many local stakeholders also get benefited from it, and in turn, it creates opportunities for the whole local community. However, our results also show that it does not reach to the poorest of poor women (extremely poor) and thus does not serve the purpose of many of its core objectives, i.e., poverty reduction of extremely poor and henceforth should not be relied upon in this perspective.

**Keywords:** microfinance; female rural entrepreneurs; household welfare; women's empowerment; difference-in-difference; rural Pakistan

---

## 1. Introduction

Microfinance can be defined as the allocating body of financial services to economically active poor who are excluded from mainstream financial service providers [1]. From the year the 1970s onwards, microfinance has been seen as a practical element in poverty reduction and policy development [2]. Microfinance is deemed to have a positive impact on the recipient's livelihood, and that, in turn, could translate into better their life quality and help the poor come out of poverty [3]. Recipients of microfinance loans seem to enjoy better socioeconomic status as reflected in their better living conditions, including better health and nutrition than non-recipients of the program [2], and reduction

in severe malnutrition is positively influenced by being a member of microfinance program for a more extended period, and it eases consumption as well. Microfinance helps recipients in better understanding the importance of health and nutrition, and that, in turn, enhances the family's food security and kids' nutrition [4]. Empowerment is generally considered as in the context of development and not just microfinance [5] and more so in case of women's empowerment.

From the 1970s, microfinance has been seen as a useful element in poverty reduction and policy development [2]. Microfinance is deemed to have a positive impact on the recipient's livelihood, and that, in turn, could better their life quality and help them come out of poverty [3]. Recipients of microfinance loans seem to enjoy better socioeconomic status as reflected in their better living condition; including better health and nutrition than non-recipients of the program [2] and reduction in severe malnutrition is positively influenced by being a member of microfinance program for a more extended period, and it eases consumption as well. Microfinance helps recipients in better understanding the importance of health and nutrition, and that, in turn, enhances family's food security and kids' nutrition [4]. Empowerment is generally considered as in the context of development and not just microfinance [5] and more so in case of women's empowerment.

Even in 21st-century women still face disadvantages worldwide in many areas, including employment, education, political representation, health, and household oppression from males because of social stigmas that upheld and encouraged such behavior [6]. Gender inequality is present in almost all countries in some form; however, the problem is more severe in low-income countries [7]. Many scholars support this assumption, who emphasize women's empowerment and given economic development [1] and prove that sustained economic development present in modern societies can be credited in parts to betterment in gender equality [8]. Even Kofi Annan (former secretary-general of UN), maintained that gender equality is a criterion for attaining other sustainable goals [9]. Although whether or not empowering women results in economic development is still an open question to debate, there seems a link between the two concepts. It seems credible that women's empowerment could change current decision-making processes and economic development [1].

In different parts of the world, many programs have been developed to achieve the better economic and social status of women, including encouraging political participation and decreasing violence against women [10]. Microfinance or microcredit is another notable approach to better women's social and economic status [11]. Although microfinance services are offered to both men and women, surprisingly, the number of women borrowers is more than men borrowers [12]. Supporters maintain that the allocation of loans and other related services through microfinance is an encouraging way of women's empowerment and achieving the goal of gender equality [6,13]. However, according to [14], female recipients of microfinance loans suffer more intimate partner violence. Advocates of microfinance assume and assert that microfinance services increase women's empowerment [6,15].

United Nations' sustainable development goals (SDG's) agenda included economic inclusion as a crucial enabler for the achievement of its goals. According to the UN, an estimated more than two billion people globally are without basic banking or financial services like bank accounts (savings), insurance, or loan products. Pakistan is one of the countries on its priority list and is home to about 5.2% of the world population with limited banking facilities. United Nations' sustainable development goals (SDG's) agenda included economic inclusion as crucial enabler for the achievement of its goals. According to the UN, an estimated more than 2 billion people globally, are without basic banking or financial services like bank accounts (savings), insurance, or loan products. Pakistan is one of the countries on its priority list and is home to about 5.2% of the world population with limited banking facilities.

Women's empowerment is reflected in all 17 of the sustainable development goals (SDGs); one such goal is to make women economically empowered. Furthermore, the 2030 agenda for sustainable development foresees a world full of universal guarantee for human rights, and that calls for the removal of all gender-based barriers, including social, legal, and economical. Our study focusses on the economic empowerment of women in Pakistan. Pakistan comes in the bottom of gender

equality, ranked at 133 out of 160 [16]. World bank studies suggest that about three billion people in the developing world can't increase their income and invest in an enterprise to improve their socio-economic status and get access to better nutrition, health, and education for their children because they don't have access to capital, savings, loans, money transfer, and microfinance. World Bank considers micro finance an effective measure to curb poverty and extreme poverty by building capacity and innovation and use its position and credibility with policymakers and regulators to make a difference to the lives of the poor [17].

Many Southeast Asian countries are agrarian, and the majority of the population lives in rural areas, Pakistan is no exception, and approximately 62% of its population lives in rural areas or villages. Their primary source of income is directly or indirectly dependent on agriculture [18]. Many South East Asian countries are agrarian, and the majority of the population lives in rural areas, Pakistan is no exception, and approximately 62% of its population lives in rural areas or villages, and their primary source of income is directly or indirectly dependent on agriculture [18]. Borrowing loans from formal lending sources like banks don not suit their profile as most of the banks charge higher interest rates and require some sort of collateral, so female entrepreneurs typically rely on informal and private unregistered lenders that charge aggravated interest rates and very harsh lending terms, which make them even more impoverished. Borrowing loans from formal lending sources like banks does not suit their profile as most of the banks charge higher interest rates and require some sort of collateral, so female entrepreneurs usually rely on informal and private unregistered lenders that charge aggravated interest rates and very harsh lending terms, which make them even poorer.

Generally, it is considered that microfinance has contributed to improving borrowers' life standards by increasing their income, savings, better access to education and health, and helping them in becoming micro-entrepreneurs, which translates into accumulation of savings, improved risk management and also help make them prepare to escape poverty. However, there is much skepticism about the poverty-reducing effect of microfinance, and critics argue that microfinance is part of the problem rather than the solution, and it pushes the poor into more poverty rather than prosperity; unfavorable terms and high-interest rates, complicated jargon are few pitfalls to name. In some extreme cases, it is alleged that it might even lead to suicides of microfinance utilizers because of the inability to repay the loans [19].

Pakistan is still primarily an agriculture-based economy that largely depends directly or indirectly on agriculture sector performance; it contributes 18.9% to the total GDP of the country and accounts for about 43% of the labor force. It is the primary source of livelihood in rural areas [18]. Female entrepreneurs are significantly disadvantaged in the male-dominated country in almost every field, including social and financial empowerment [17]. Pakistan is ranked 151 out of 153 countries according to the latest UN gender equality ranking [20].

From the period of 2007–2008, the food price crisis-affected the welfare of the poor, including women all over the world [21]. Likewise, the slow growth in the rural economy of Pakistani is a result of the increased supply of rural credit below the demand of total borrowers [22]. In general, empirical evidence indicates that gender-specific responses resulted following structural adjustment and Asian economic crisis in particular, which show women to be disproportionately affected by shocks from the period 2007–2008 food price crisis, suggests on female producers and consumers could be similar to those of the previous crisis [23]. In consideration of the above statements, consumption price and production price risks are global phenomena; henceforth, efforts are made to include inflation-adjusted values according to national average inflation values.

The present study was carried out to analyze the role and contribution of MFI's (microfinance institutes) in women's economic empowerment using income and consumption as welfare indicators. The difference in difference method is applied to investigate the effects of microfinance on its recipients. The impact of the microfinance program is measured through the consumption and income of households by comparing participants and non-participants.

## 2. Literature Review

A lot of empirical studies from Bangladesh support the idea of women's empowerment through effective use of microfinance and support the notion that microfinance helps in building and improving the following skillsets of female entrepreneurs. (i) decision making and business insight; (ii) social and financial empowerment (iii) more robust contribution to kids' schooling, nutrition, and health. Furthermore, microfinance institutions' term women as more creditworthy than their male counterparts and thus improving microfinance institutions financial viability and sustainability [24].

As microfinance enhances women's income-generating capacity, it makes them less dependent on their husbands for money [25]; and they can easily pay for things like food, child's education, and paying for healthcare [26]. In one cross country study, it is found that microfinance institutions produce positive impacts at the macro level and helps in reducing severe poverty, this impact is more notable in countries with more significant portfolios of microfinance loans per capita [27]. Increase in women earning power can give them more bargaining power in troubled and abusive marriages [28], and it could also help in the reduction of violent domestic behavior by husbands [29]; the association between low income/poverty and domestic violence is widely recognized by many scholars [28].

Microfinance exists to help those individuals and microenterprises which are unable to avail services of leading stream financial service providers due to a variety of reasons including; lack of initial capital, lack of knowledge and skills, lack of collateral or being more risky clients and henceforth, provision of loan products to poor low income generating individuals, by microfinance institutions, helps in creating income and employment opportunities for poor households and communities [30].

However, the empirical literature regarding microfinance true impact in eradicating poverty and equipping poor with tools to free themselves from vicious circle of poverty is full of controversies; making the welfare evidence of microfinance institutions questionable, with advocates [24] provide ample evidence of its beneficial socio-economic impact, presenting it as panacea for the poor; while skeptics and critics stating completely otherwise [31]. Some authors even reported zero effect of microfinance on welfare [32], while some other authors also reported mixed impacts of microfinance on its recipients welfare [33].

Microfinance has experienced fast and successful growth in developing countries in recent decades. Microfinance is defined as a provider of financial services such as deposits, credits payment services to the poor, and low-income households and their microenterprises [21]. Microfinance institutions focus on providing loans to millions of disadvantaged and excluded sections of society (both men and women) and help them establish small and microenterprises to generate income [22]. It is due to this positive effect of microfinance and its contribution to the economy; that encouraged and motivated many scholars to carry out studies to know its impact, outreach, and sustainability. [23–25].

In 2000, according to the state of the Microcredit Summit Campaign 2001 report, nearly 74% of the 19.3 million of the world's poorest women now had access to financial services through microfinance institutions. Most of these women have access to credit to invest in a business that they own and operate. The vast majority of them have excellent repayment records, despite the day to day issues that they face. It shows that women are considered better clients as compared to men. Microfinance institutions priority for female clients can be summarised as follows.

1. Economic Development

Research done by the world's many leading organizations like UNDP, UNIFEM, and World Bank suggests that gender inequalities hinder economic growth, adds to poverty, and lower socioeconomic status of masses [26].

2. Poverty alleviation

Contributes to women's development strategies by giving them access to capital, thus making them part of economically productive segments of society and help them escape poverty [27].

3. Women's empowerment

There are more poor women in the world than men. According to the World Bank's gender statistic database, women have a lower employment rate than men in most of the countries [28]. This

rationale supports women's access to financial services, as it clearly shows that women are relatively more disadvantaged than men. By providing financial services for income-generating activities, for women entrepreneurs, microfinance institutions can significantly reduce women's vulnerability to poverty. Women generally spend most of their income on their families. Therefore an increase in their income translates into an increase in family welfare. A study conducted by Women's Entrepreneurship Development Trust Fund (WEDTF) in Zanzibar, Tanzania, reported that 55% of women's used their increased income to purchase household items, 18% goes to spending on schooling, and 15% is spent on clothing. Therefore, assisting women in income-generating activities through the use of microfinance will generate a multiplier effect that enlarges the impact, not just for the borrower but for the whole family [29].

4. Efficiency and sustainability

Better loan repayment records are one of the many reasons for MFIs more focus on women borrowers. It has emerged that women's repayment rate is typically far superior to those of men. Lower default loan rates is an essential prerequisite for a smooth and sustainable operation of any financial institution [29].

Below are different ways in which women empowerment can be substantiated

1. Decision making

Microfinance programs for women borrowers are found to be useful in making them deciside freely without any coercion. [31].

2. Self-confidence

Self-confidence is one of the most crucial areas of change for empowerment, but it is not easy to measure. Confidence is a complex concept relating to both women's perception of their capabilities and their actual level of skills and abilities. The use of microfinance programs has been found to affect this trait positively, and it has resulted in bettering women's ability to resist domestic violence and encouraged them to demand fair minimum wages [32].

3. Improved status in the home

It helps women participation in income-generating activities and the resulted better financial standing is assumed to strengthen women's bargaining position within the household, thereby allowing them to assert themselves more effectively in strategic decision making [33].

4. Family relationships and domestic violence

It contributes significantly to women's financial freedom, and that translates into being more asserting voice at home. It also helps in reducing domestic violence against women [34].

5. Women's involvement and status in the community

Microfinance contribution to women financial resources greatly influences women's bargaining power in the family, and it improves women status in the community as demonstrated by a study carried out in Tanzania [35].

The above literature clearly shows a link between the use of microfinance and women empowerment and hence the reason for our study to know the exact impact of microfinance on women borrowers in Pakistan's Punjab province. Additionally, we would also like to evaluate the notion of serving the poorest of the poor women by microfinance entities.

Based on the above literature the following conceptual model was developed that represents diagrammatic representation of the framework that shows suggested relationships and intermediating aspects in our study represented in Figure 1.

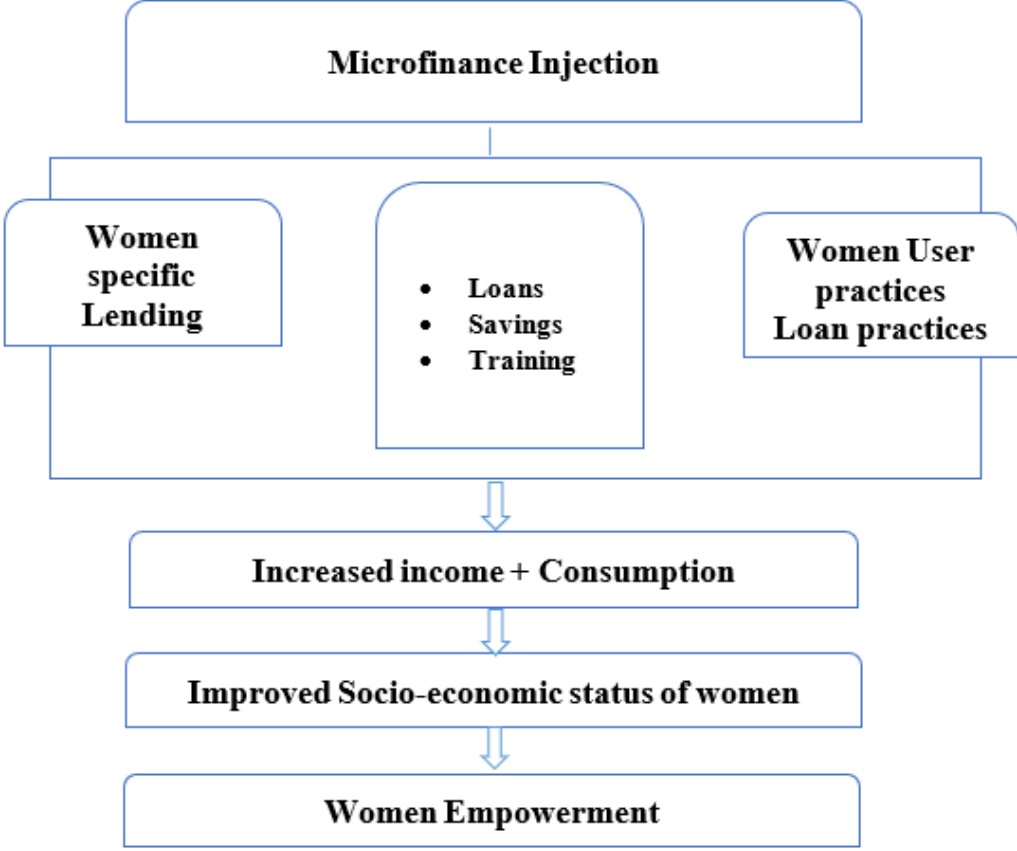

**Figure 1.** Conceptual Framework.

## 3. Data Collection

### 3.1. Survey Area

A survey was conducted between January 2017, March 2017 in Pakistan's punjab province which is most populous, most literate and economically more productive region as compared to other provinces of the country [36].

Traditionally the area belongs to male dominated society, but socio-economic development of the country, also helped in promoting female employment and entrepreneurial abilities [18,37].

### 3.2. Data Sources and Data Description

Both primary and secondary data was collected through well-structured questionnaire through household/ workplace survey. A list of borrowers was collected from respective regional branches of banks. Recipients of loan belong to the country's leading microfinance banks; APNA Microfinance Bank (AMFB), FINCA Microfinance Bank, Khushhali Microfinance Bank, National Rural Support Programme Microfinance Bank (NRSP Bank), and Microfinance Practitioners Pakistan (Microfinance connect). In this study, the participant group is the household where female member of the household took microfinance loan and the non-participant group is the household that never took microfinance during the survey time.

The earliest year in which the program was implemented was 2011; therefore, we exclude the year 2010, which is soon after and instead use the year 2009 for empirical data collection, and the year 2016 is chosen as program post period. A two-year household panel data set is used (2009/2016) to estimate the difference-in-difference (DD) model for impact on welfare, using income and consumption as measures of welfare indicators, while accumulative loan amounts show household participation in the microfinance program, which is a predictor/independent variable in our study of impact analysis.

As in the previous study by 50 stated that earning immediately before the program could be biased upward due to a trend called "Ashenfelter dip". Therefore to overcome this problem, while applying the DD method, it is essential to use the program pre-period early enough [36].

### 3.3. Sample Selection

Two groups of households are used in our study one borrowers and other non-borrowers, and a multi-stage stratified random sampling technique was used to get a household sample. In the first stage, townships were selected that the program has access to. A total of 13 such municipalities were selected. The microfinance program has been functioning since 2011 in 4 out of 13 towns. Subsequently, sample villages were chosen from sample townships. A total of 4 communities were taken from each district, and altogether 50 villages were selected. Based on the borrowers lists available, from each regional branch office in the selected districts (Lahore, Gujrawnala, Sialkot and Faisalabad) from the following microfinance institutions (APNA Microfinance Bank (AMFB), FINCA Microfinance Bank, Khushhali Microfinance Bank, National Rural Support Program Microfinance Bank (NRSP Bank), Microfinance Practitioners Pakistan (Microfinance connect); a total of 350 microfinance female clients were randomly selected to be interviewed by female staff, female interviewers were used to interview and collect data, keeping in view local traditions. Another 130 households were selected from village committee head (locally called namberdar), who did not take out any loan. Due to various reasons; refusal to participate in survey, non-availability and incomplete survey responses, 50 clients were dropped from borrowing group and 30 were dropped from non-borrowing group, henceforth bringing the final sample to 300 borrowers and 100 non borrowers. Borrowers female were defined as those who took micro loans from the above stated microfinance institutions in above said districts in years (2009/2016) and non-borrowing households were defined as the ones who never took loan but belonged to same district. It was made clear to the respondents that participation is totally voluntary, and respondents were ensured of complete confidentiality and anonymity of their responses. Respondents were made aware of the purpose of survey and a verbal consent was obtained, prior to start of data collection.

## 4. Methodology to Impact Assessment

This study aims to analyze the role and contribution of MFI's in women economic empowerment using income and consumption as welfare indicators. Microsoft Excel 2017 and Stata/MP 14.0 for windows were used for data analysis.

### 4.1. Analytical Framework

In this study, the impact of the microfinance program is measured through consumption and income of households by comparing participants and non-participants.

Suppose, wj is a binary indicator that is equal to 1 for those who participate and 0 for those who do not participate. Furthermore, suppose Yj1 represents the outcome of interest in household j participation in the microfinance program, and Yj0 represents the same outcome for non-participation. So the result of microfinance can be find out using the following formula [36,38].

$$\Delta_j = Y_{j1} - Y_{j0} \tag{1}$$

However, to get the real impact of microfinance programs, we must resolve two-issue; first, i) unobservability and ii) missing data. As the same household cannot be analyzed simultaneously for both being participant and non-participant, this gives rise to the issue of missing data, and that in turn hinders our way of finding the correct impact of microfinance. As suggested by 38 and 39, one or another factor of the difference is absent from Equation (1).

This problem is solved by applying group statistics, e.g., 'average effect of treatment on the treated (ATT). Following that, the correct impact of the program can be find out using following ATT equation [36,39]:

$$\gamma = E(Y_{j1}| w_j = 1) - E(Y_{j0} | w_j = 1) \tag{2}$$

Counterfactuals are used to address this problem by using the framework of treatment/control. In this case, a group of non-participants of the program is chosen as a control group, and readings of the group act like 'counterfactuals' participants of the program (treatment group). The real impact of the program is then calculated using the following method:

$$\gamma^* = E(Y_{j1} | w_j = 1) - E(Y_{k0} | w_k = 0) \; (j \neq j \in N) \tag{3}$$

Where $\gamma$ = estimation of y. J and k = two different household, where j = participant and k = non-participant

yj1 = outcome investigation j. $Y_k0$=same outcome investigation of household k [40].

In this study, we are comparing the average welfare impact on households, utilizing microfinance (income and consumption) among borrowing households (treatment group) and non-borrowing households (control group).

### 4.2. Empirical Model and Estimation Strategy

Annual household income and annual household consumption are used as welfare indicators. Household income is the sum of all incomes earned through agriculture and non-agriculture, wages, and self-employment by all household members. Total consumption of the household is the sum of all food and nonfood consumption.

After careful evaluation of literature, it has been found out that the issue of selection bias must be carefully addressed to make the results more reliable and accurate; the question of selection bias arises when microfinance participants are exposed to unmeasured dynamics, these unobserved factors or dynamics affect the outcome of impact. According to [40–42], two primary sources such bias are non-random program placement, which occurs by unaccounted village attributes and self-selection of households into the program. Therefore it was argued that comparison of borrowing and non-borrowing without taking into account heterogeneity (e.g., unobserved characteristics of the study population) and that would ruin the whole impact outcome of the program, as this may have risen at least partly from those unaccounted or uncontrolled factors.

Panel data is used to moderate the issue of selection bias in our impact estimation. The use of the difference in difference method or DD is an effective way of tackling such a scenario in social sciences where data is not purely experimental. This requires observation of two groups over two-time durations.

Here the first group, participant group consisting of recipients of the program (i.e., post-program periods) but not before the launch of program (or pre-program period) and second group non-borrowing group consists of households who didn't get loan funding during either of the periods [43,44]. Following 61 and 62 the DD (standard method) can be shown as follows:

$$Y_{jt} = \beta_0 + \delta_0 d2_t + \beta_1 P_j + \gamma M_{jt} + \varepsilon_{jt} \tag{4}$$

Where Yjt = natural logarithm of household investigated outcome (household consumption or income) for i household at period t; d2t being dummy variable = 1 for t = 2, i.e. (post-program period) and 0 for t = 1 (preprogram period). Here pi is a group dummy variable and takes the following values one for loan recipient household and 0 for non-borrowing house holds.

$M_{jt}$ is an interaction term of the product of $d2_t$ and $P_j$ and takes the following values 1 for program participants and 0 for non-borrowing.

While j0 estimates time influence, observed by both control and treatment group, 1 estimates the potential difference in time-variant on averages overall between two groups, and it is the main parameter that measures interest on average borrowing group (treatment group). At the same time, Ejt is an idiosyncratic error for households and time having zero value at all periods.

The critical assumption of the standard DD method, also called the common trend assumption, is that Y would be zero in the absence of the program, or $E[\varepsilon it|Mit] = 0$. In other words, the average change in the outcome variables (Yjt) would not have been systematically different between the borrowing group and the non-borrowing group if there were no program [45,46]. Under this assumption, an unbiased estimate of y can be obtained using the following equation:

$$\hat{Y}_{sdd} = \Delta\overline{Y}b - \Delta\overline{Y}n = E(Y_{j,t=2} - Y_{j,t=1} \mid P_j = 1) - E(Y_{j,t=2} - Y_{j,t=1} \mid P_j = 0) \tag{5}$$

Here it is important to note that the standard DD method assumption, also called trend assumption, is that in the absence of program or $E[\varepsilon jt|Mjt] = 0$.

According to [45,46], the dependent or outcome variable Yit will not come out significantly different among borrowers and non-borrowing, so keeping in view this assumption an unbiased estimate of y can be obtained as follows:

Where delta denotes change as t = 1 and t = 2. SDD = standard DD estimator, and the overbar is for averages among households. B and N represent borrowing and non-borrowing households. By subtracting average differences in group of borrowing in Equation (5), two types of DD estimation bias are created and needs to be removed. Bias due to comparisons between cross-sectional groups in post-program period, that could arise because of permanent differences in the two groups and is captured by beta1, that is unrelated to program and similarly the resulting bias due to two periods of borrowing group that could have been caused by time trend captured by i0 and irrelevant to the program [47–49].

As reported by 45, the standard DD method is valid only if the household selection method is random (like in experimental settings) time and fixed group assets conditional. Balancing is done through randomization over time, between these two groups, the pre-program attributes linked with outcome variable/dependent variable (Yjt). However, since our study is non-experimental in nature, both households may be systematically different and unbalanced in case of pre-programming features that could be related to the dependent/outcome variable. Therefore, change patterns could be significantly different if no loan programs were paving the way for a biased estimate of the impact of the program [40,46,47]. To improve the resulting efficiency, we further use the adjusted DD method to estimate the program impact. We first include a vector of observable household characteristics in the regression as control variables to adjust for the observed differences between the two groups. Additionally, the fixed effects method is applied to control for the unmeasured household and village attributes. Other studies by 56 and 66, household fixed effects estimation, which treats the household-specific component of the error as a parameter to be estimated, can resolve selection bias at both household and village level, including the upper level (e.g., township-level), based on the assumption that the unobserved factors at household and village level are invariant over time. Accordingly, our study employs household fixed effects estimation to correct for the potential bias. The following two-way fixed effects model illustrates the econometric model:

An adjusted DD method is employed to improve efficiency results by including a vector of characteristics of household observations in the regression as a control variable to fine-tune observed differences among the two groups. Furthermore, the fixed effect technique is used to control unmeasured household and village factors, as could be seen from studies by 56 and 46. Consequently, we use fixed effects household estimation to rectify potential bias, and the econometric model evolved into the following form:

$$Y_{jt} = \beta_o + \delta_o d2_t + \eth X_{jt} + \gamma M_{jt} + h_j + U_{jt} \tag{6}$$

Here Yjt, d2t, and δo are defined similarly as in Equation (5), and xjt shows household characteristics vector (like family size, age). At the same time, Mit is the treatment variable and is the main component of microfinance impact on loan recipient households, and hi is household fixed effects capture of households of the time constant that affect Yit. At the same time, uit is the idiosyncratic error that

changes over time and shows unobserved household characteristics that evolve. It is important to note that the unit is normalized to always have zero for each period and is considered to be independent of program variable Mjt and shows the same distribution in different intervals of time, so unit γMjt, but still correlated to hj [40,47]. According to 64 and 62, by adding Xjt as repressors could help control for confounding trends and better estimate efficiency by way of reduction in residual variance.

In the previous study by 56 and 52, the adjusted DD method based on fixed regression eased the strict restriction i.e., randomization through standard DD method allows households, two groups, to be systematically different. This results in an adjusted DD method being the same in both periods, thus evaluating differences among the two groups of households are free of the bias and results in actual program effect. Table 1 below shows main variable used in our study

**Table 1.** Variables description.

| HHAI | Continue | Log (Household Annual Income) |
|---|---|---|
| HHAC | Continue | Log (household annual consumption) |
| AGE | Continue | Age (Borrower) |
| SCLEDU | Continue | Kids (school going age) |
| HHSZE | Continue | Size (household) |
| HHERNRS | Continue | Total household earners |
| YEAR | Dummy | Year indicator (1 for 2017, 0 otherwise) |
| PARCPN | Dummy | Participation indicator (1 for yes, 0 otherwise) |
| Cumul continue | | Cumulative loan borrowed by household |

Source: Survey results. HHAI = total household annual income consumption; AGE = age of borrower in years; SCLEDU = total number of kids in school going age; HHSZE = total size of household; HHERNRS = total number of earners in household; YEAR = indicates year; PARCPN = measures participant and non-participant, 1 shows program participant and 0 shows non-participant; Cumul = sum of total money borrowed by household.

## 5. Empirical Results and Finding Discussions

### 5.1. Characteristics of Micro Loans

The General characteristics of micro finance borrower is given in Table 2.

With regards to maximum and minimum borrowing amounts, the majority (38%) take out loans between 50,000–100,000, followed by (32%) who borrowed up to 50,000. Few borrowers take out larger loans of (100k+ and 150k+), 22%, and 12%, respectively.

The majority of the loan received is for short term i.e., less than a year (63%). The primary purpose of the loan was for non-agriculture purposes, including but not limited to a beauty parlour, handicraft making, tailoress livestock raising, and poultry farming, etc.

Majority about (88%) used their loans for non-agricultural purposes and only a small 12% for agriculture purposes. The majority of borrowers preferred to repay loans monthly (55%). The average total borrowing amount is Rs.39359, while the average earning is Rs.48256.The loans that are used for other purposes than agriculture the repayment period is ordinarily short as these activities can produce income more frequently, and this finding is in line with the result of 54 and 55.

### 5.2. Impact Estimation with Standard Difference-in-Difference Method

In a standard DD, method see Equation (5); the treatment variable i.e., Mjt (microfinance), is a binary form and takes values of 1 for program participation and 0 for non-participation. Furthermore, the assessed model is a logarithmic function, and here the dependent/outcome variable is the natural logarithm of consumption/income, so the coefficient y of treatment variable measures the estimate average percentage difference in the treatment variable when multiplied by 100 [50]. Please see Table 3 for details. It is clear from our results that welfare measured through annual household income and yearly consumption household has significantly improved as compared to non-borrowing households in the years between 2010 and 2016 (see Table 3).

**Table 2.** General characteristics of micro finance borrowers (N = 300).

|  | Subtotal 300 | % to N 100 |
|---|---|---|
| Loan type—individual | | |
| Loan amount (single) | | |
| <50,000 | 95 | 32 |
| 50,000–100,000 | 115 | 38 |
| 100,001–150,000 | 65 | 22 |
| >150,000 | 35 | 12 |
| Cumulative loan amount = Rs.39,359 | | |
| Loan term | | |
| Short term (<1 year) | 190 | 63 |
| Long term (1–3 years) | 110 | 37 |
| Collateral | | |
| Yes | 40 | 13 |
| No | 260 | 87 |
| Total | 300 | 100 |
| Frequency (payment) | | |
| Monthly | 165 | 55 |
| Bi-annually | 95 | 32 |
| Annual | 40 | 13 |
| Total | 300 | |
| Purpose of loan | | |
| Agricultural | 35 | 12 |
| Non-agricultural | 265 | 88 |
| Total | 300 | |

Source: Survey results.

**Table 3.** Standard difference in difference valuation of household (HH) welfare.

|  | Year | Year | Difference | Year | Year | Difference | DD Impact Estimator |
|---|---|---|---|---|---|---|---|
|  | 2010 | 2016 | (2010–2016) | 2010 | 2016 | (2010–2016) |  |
|  | (1) | (2) | (3) | (4) | (5) | (6) | (7) |
|  | Yb 10 | Yb 16 | D1 = (Yb16–Yb10) | Yn 10 | Yn 16 | D2 = Yn16–Yn10 | Ysd = D1–D2 |
| HHAI | 3.894 (0.016) | 4.189 (0.015) | 0.295 *** (0.024) | 3.861 (0.045) | 4.090 (0.043) | 0.229 *** (0.056) | 0.066 * (.032) |
| HHAC | 3.628 (0.019) | 3.874 (0.016) | 0.246 *** (0.025) | 3.674 (0.036) | 3.889 (0.034) | 0.215 *** (0.051) | 0.031 (0.25) |

Source: Survey results. Note: Entries represent means of log household annual income and log household annual consumption for the borrowing group and non-borrowing group, respectively; Numbers in parentheses are standard errors. * Represents 10% significant level for the t-test. *** Represents 1% significant level for the t-test.

Our results showed significant and substantial improvement for both annual household consumption (HHAC) and annual household income (HHAI), HHAI is increased approximately 29% over a period of 6 years, and it's statistically significant at 1%, and HHAC also shows positive change during this period. However, as discussed before, this could be due to time influence and microfinance programs. So, to get the actual effect and this factor, potential time influence must be accounted for. Column 7 in Table 3 shows the average outcome for non-borrowing groups.

Standard DD estimation is arrived after differencing the mean gains between two groups (see column 6 Table 3.) So, the average household annual income (HHAI) shows an increase of 6.6% due to the direct impact of program participation and is significant at 10% statistically. However,

annual household consumption (HHAC) is increased by approximately 3.1% but is not statistically significant. (Table 3, column 8).

Although the standard DD method shows positive and statistically significant effect marginally; the impact on household consumption is not statistically significant, according to standard DD method, it is assumed that only treatment variable and no other variable could have brought such change in investigated outcome (Yjt). However, this would be only true if the two groups are similar in household characteristics. Henceforth, the standard DD method would lead us to biased estimation of impact without controlling for other variables.

### 5.3. Impact Estimation with Adjusted Difference-in-Difference Method

Henceforth use of adjusted DD method to exclude potential deficiency of standard DD method to get an impact on welfare based on fixed effects regression is presented in Equation (6).

Table 4 below shows that participants of the microfinance program generally increases by 5.19% and annual consumption by 5.30% in contrast to non-borrowing. The results showed that the adjusted DD method differs only slightly from those of the standard DD method. This led us to the conclusion that borrowing and non-borrowing households do not influence the trend investigated significantly [51].

**Table 4.** Adjusted diference difference (DD) estimation of the impact of program participation. Outcome variables.

| Intercept | HHAI | HHAC |
|---|---|---|
| Intercept | 4.360 (0.4569) | 4.253 (0.4239) |
| Year | 0.1693 (0.0630) | 0.1435 (0.0630) |
| Control variables (xit) | | |
| AGE | 0.0003 (0.0001) | 0.0003 (0.0001) |
| SCLEDU | 0.0159 (0.0249) | 0.0056 (0.0215) |
| HHSZE | 0.0004 (0.0856) | −0.0283 (0.0809) |
| HHERNRS | 0.0493 (0.0235) | 0.0263 (0.0284) |
| Treatment variable (Mit) | | |
| PARCPN (binary) | | |
| F Statiscs (424.319) | 5.19 | 5.30 |
| Household fixed effects | jointly significant | jointly significant |
| $R^2$ | 0.8654 | 0.8603 |
| Total observations | 900 | 900 |

Source: Survey results.

So the use of aggregate borrowing amounts by households is a much better way of their involvement in our microfinance program. Mjt in eq6 describes a continuous variable showing loan size and evaluates the additional amount of loan borrowed by households [39,52].

Here we see that the explanatory power of the model of fixed effects is satisfactory (R2 in Tables 4 and 5). Furthermore, f test is also conducted to get the impact of coefficients of household fixed effects (hj) are zero to test the null hypothesis. F statistics show significant at a 1% level in both tables. So we strongly reject the null hypothesis in approval of the fixed effects model for rectifying impact estimation in selection bias, as is evident from Table 5, that microfinance has a positive impact on household welfare. On average, every Rs.1000 raises annual household income by 0.26% and annual consumption by 0.21%.

The average loan by the average household is Rs.39,359 (see Table 2); the results show that the recipients of loans enhance their annual income and consumption by 10.23% and 8.26% approximately, respectively, as compared to non-borrowers. Thus we can conclude that borrowers will benefit more when they take out more loans, and this finding is consistent with the result of 40 and 51.

**Table 5.** Adjusted DD estimation of the impact of micro loans.

|  | HHAI | HHAC |
|---|---|---|
| Intercept | 4.2695 (0.4439) | 4.2013 (0.4023) |
| Year | 0.1314 (0.0563) | 0.1048 (0.0542) |
| Control variables (xit) |  |  |
| AGE | 0.0001 (0.0001) | 0.0001 (0.0001) |
| SCLEDU | 0.0169 (0.0225) | 0.0089 (0.0204) |
| HHSZE | −0.01690 (0.0844) | −0.0453 (0.0269) |
| HHERNRS | 0.0614 (0.0359) | 0.0350 (0.0269) |
| Treatment variable (Mit) |  |  |
| Cumulative in Rs.1000 | 0.0026 (0.0003) | 0.0021 (0.0004) |
| F statistics | 5.69 *** | 5.75 *** |
| Household fixed effects | jointly significant | jointly significant |
| $R^2$ | 0.8891 | 0.8875 |
| Total observations | 900 | 900 |

Outcome variable Source: Survey results.

Our empirical outcomes, as based on the DD adjusted method, shows that microfinance leads to the betterment of household annual income and consumption. Through impact for binary is disappointing and is not statistically significant for any of the dependent variables investigated; however, when used cumulative borrowings, the welfare impacts estimated resulted statistically significant at 1%. Although results show that the effect of microfinance on welfare as substantial statistically, but it is not evident in the economic sense. The following values in Table 5 will make this point clear, the coefficient (y), variable commutative borrowings are very small in income and consumption as compared to coefficients of variables like earnings. So when multiplied by 100, this gives us minimal values in percentage and shows the insignificant or minimal effect on the welfare of borrowing household's income and consumption [51].

### 5.4. Poverty Targeting of Microfinance Program

In our study, a household is considered poor if his income falls below $1 per day, as defined by the Ministry of finance Pakistan [53]. To evaluate program effects, two lines of poverty were used the low poverty line and low-income line to find out about the poverty status of the household.

Table 6 showed, poverty targeting of the microfinance program. The upper panel shows poverty across the post-program period i.e., 2009; it shows little or slight poverty incidence. An only a small percentage of the borrowing household is categorized as poor with either poverty line, which leads us to the conclusion that the program does not primarily focus on poor exclusively. Using the poverty status of post-program households can result in a biased or inaccurate evaluation of program purpose. As households can get microfinance loans and enhance their consumption and income and leave the poverty line, henceforth poverty status of households in 2009 i.e., the post-program period is also examined, and the results are shown in the lower panel in Table 6. The results do not show significant changes as compared to those of the 2016 post-program period, for example, the poor account for 4.3% of borrowing and 14% for households without financing. So this leads us to the conclusion that the majority of the program participants were not poor when they take out loans [39].

**Table 6.** Poverty status of the sampled household by type and year.

| | Borrowing Household | | Non Borrowing Households | | Total | |
|---|---|---|---|---|---|---|
| | Count | Share % | Count | Share % | Count | Share % |
| Poor by PL 2016[x] | 9 | 3.0 | 6 | 6.0 | 15 | 3.75 |
| Poor by IL 2016[y] | 8 | 2.66 | 9 | 9.0 | 17 | 4.25 |
| Sample size | 300 | | 100 | | | 400 |
| Poor by PL 2009[x] | 13 | 4.33 | 14 | 14.0 | 27 | 6.75 |
| Poor by IL 2009[y] | 12 | 4.0 | 13 | 3.0 | 25 | 6.25 |
| Sample size | 300 | | 100 | | | 400 |

Source: Survey results. Note: x represent the low poverty line and is less than $1 for both 2009 and 2016 and y represents the low income level is less than $1.50 for both 2009 and 2016 in above Table 6. Despite microfinance contribution to income and economic growth of the borrower, the results showed that recipients of the microfinance program are not the poorest of the poor.

## 6. Conclusions

This study empirically examines the role and impact of microfinance on Pakistani rural female entrepreneurs, by recording total income and consumption through means of difference in difference method. Our results are in line with other major findings and studies that support the idea, that microfinance increases household consumption and income and that, ultimately leads to welfare improvement of recipients. Furthermore, it shows that as borrowers get into higher loan cycle they are more likely to reap benefits of microfinance, and this idea is supported by growing loan size. But the program benefit may have been underestimated due to nonrandom program-bias, in other words the program benefits could have also been reached to control group as well (non-borrowers). Henceforth, the true impact of microfinance program may be much stronger than as shown by our findings. Therefore, it does not just benefits only borrowing group rather many stack holders are benefited and in fact creates economic and business opportunities for the whole local community and not just borrowers.

However, as our results indicate that the main beneficiaries of the program are not the poorest of the poor women, and it suggests that those benefit more from the program who have better financial standing and ability to invest in microenterprises to generate income, say, micro enterprises. Henceforth, it seems out of the reach of extremely poor due to lack of income generating skills, knowledge, inadequate capital or poor repayment capacity. Therefore its prudent to say that microfinance is not well suited for the poorest of poor and government should adopt other policies to help them out of vicious circle of poverty; e.g., grants or interest free or highly subsidized loan products together with basic business running skills package to better equip them for proper utilization of loan before they could apply for microfinance. In the meantime, basic necessities of life for extremely poor, like health, education, food and shelter should be met by government support programs.

Limitations: As the study is based only on small sample size and represents only few rural and semi-rural areas of Pakistan's Punjab province, it does not represent the true picture of whole country, henceforth further studies on larger scale are required to validate and cement the findings. Furthermore, future research can be carried out using different or additional welfare indicators other than just income and consumption.

Recommendations: Based on our findings, it is prudent to say that government should help out poorest of the poor women (extremely poor), through other means (interest free loans, grants and social security measures like free health insurance and social security), as microfinance seems beyond their reach. Furthermore, basic business running skills should be provided highly subsidized or free of cost to poor future aspiring women entrepreneurs.

**Author Contributions:** The research was conceptualized by T.A.K. and S.J.; methodology was refined and organized by T.A.K. and F.A.K.; data collection and analysis was carried out by T.A.K. and Q.V. The introduction and review Sections were done by T.A.K. and Q.V.; the first draft was prepared by T.A.K..; and shared with S.J. for

valuable comments. The final revisions were made by T.A.K., I.A. and S.J. All authors have read and agreed to the published version of the manuscript.

**Funding:** This research did not receive any specific grant from funding agencies in the public, commercial, or not-for-profit sectors.

**Acknowledgments:** Authors would like to pay thanks to the Huazhong Agriculture University Scholarship Department, without this support, it was impossible for the first author to do his doctoral study. As the study is the part of the first author's doctoral dissertation so there is no specific funding of the research, except for a doctoral scholarship from Huazhong Agriculture University.

**Conflicts of Interest:** The authors who are listed immediately below the title of the manuscript certify that they have NO affiliations with or involvement in any organization or entity with any financial interest (such as honoraria; educational grants; participation in speakers' bureaus; membership, employment, consultancies, stock ownership, or other equity interest; and expert testimony or patent-licensing arrangements), or non-financial interest (such as personal or professional relationships, affiliations, knowledge or beliefs) in the subject matter or materials discussed in this manuscript.

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
