# Peer review of "Microfinance Facility for Rural Women Entrepreneurs in Pakistan: An Empirical Analysis"

_agriculture, doi:10.3390/agriculture10030054_

Round 1

Reviewer 1 Report

This is a good study that provides some interesting data on the issue of whether microfinance borrowing is beneficial or not, and to what extent. The methodology appears reasonable, with adequate efforts to adjust for the inevitable statistical problems of a non-randomized non-experimental dataset (though a little more discussion on causality would be in order, as noted below).

What would be needed for revisions are:

A brief literature review that recognizes other relevant empirical research (rather than asserting it does not exist) and sets up whether the authors are testing specific hypotheses or investigating whether the data supports proponents or sceptics of microfinance; Taking into account comments below; Copy-editing by a native English speaker – although the writing is generally clear and understandable, the flow and grammar need some attention.

ABSTRACT

It is not correct that “no recent study assesses microfinance effects on rural female agricultural entrepreneurs.” A quick Google search finds, among others:

“Rural women’s access to financial services: Credit, savings and insurance,” Diana Fletschner and Lisa Kenney, FAO, ESA Working Paper No. 11-07, 2011 “The effect of microfinance on the empowerment of women and its societal consequences: A study of women self-help-group members in Andhra Pradesh”, KNUT-ERLAND BERGLUND, Department of Economic History, Uppsala University,  Minor Field Studies-reports 2007:1 Effects of Microfinance on Agricultural Occupation: Case Study in Battambang Province, Svay Sopheana et al., DOI: 10.7763/IPEDR. 2012. V46. 13

Also, since less than 12% of the borrowers were in agricultural, wouldn’t it be better just to say “rural female entrepreneurs” and drop the “agricultural”.

Although the substantial bibliography lists a number of publications on microfinance and on women’s empowerment, these are only referenced in the Introduction – which does an adequate job of summarizing arguments that microfinance can help address poverty and empowerment and of noting that other studies have questioned that premise or found negative effects. There needs to be a Literature Review section that reviews more specifically what studies of microfinance and women entrepreneurs (in agriculture or MSMEs) have investigated and concluded, in order to set up what the authors would expect to find in the context being studied – or to state explicitly that they are going to investigate whether the positive or negative effect hypothesis holds in Pakistan (for the sample group).

Use of data at a point in time to establish a control group raises a fundamental problem of causality. It is noted that the “same household cannot be analyzed simultaneously for both being participant and non-participant”  -- i.e., there is no before-and-after data for the treatment group. So non-participants are used to serve as a control group who did not receive the “treatment.” It is assumed that microfinance is the “treatment.” But one could just as easily view higher income as the “treatment” – in which case we are testing the reasonable hypothesis that higher income yields better access to microfinance. [However, one could also argue that, at some point, people with still higher incomes don’t need microfinance, because they can access banks.]

The authors need to be careful to admit that what is being observed is an association, and that causality could be in the other direction. But based on the literature and a conceptual framework, it is presumed that if higher income/consumption is observed, it would be an effect rather than a determinant of participating in microfinance. And, further, the difference-in-difference methodology helps to sort out the direction of causality.

Some studies handle this problem by taking as the control group people who have joined the microfinance organization but have not yet received loans – thus they can be presumed to be comparable in other characteristics – including level of income/consumption at the time of entry to the organization -- to other members who have previously received loans.

This study uses the next-best approach, which is to observe both partipants and non-participants at two points in time, and to attribute any differences in their relative changes over time to the treatment variable. This is adequately explained on lines 158-162; and the flaw in the approach is recognized in lines 180-4, followed by a reasonable effort to deal with it.

However, it appears that the study does not actually gather data at two points in time. Rather, the ‘treatment group’ consists of participants who had a loan at the time of the survey but were not participants in the “pre-program period” (line 144).  Lines 240-2 clarify that 2009 was chosen to represent the “pre-program period” (to minimize any anticipatory effects in the lead-up to the 2011 launch) and 2016 for post-program. But doesn’t that mean that people were being asked to recall their consumption and incomes 8 years prior to the 2017 survey?!? (since there was no baseline survey). How reliable can recall data be for that previous period (especially for income, which is notoriously difficult to capture adequately through self-reported survey responses)? We need more detail on how income and consumption were measured, and the uncertainty involved if they were indeed based on recall data.

Also, assuming these were monetary values, how was inflation handled?

Also, we should know the basis on which people were accepted into the program. As noted above, it may be that those who qualified for the program already had a higher level of income, education, skills, etc. (which can only partially be controlled for through the regression technique for the adjusted DD approach.

Line 89:  Is it an established fact that microfinance per se caused the suicides? It may be more prudent to say “alleged” if there was simply an association between suicide and being a microfinance client (among other factors). [In any case, it was the inability to repay loans (due to various factors) that is involved, not the availability of loans, savings etc. as such.

Lines 309-310:  Using loan size as a continuous variable for usage of microfinance instead of a dummy variable for participation is a useful innovation, with good results.

Lines 333-4 usefully clarify that, even though a statistically significant difference may be found, the magnitude of the difference is relatively small – and thus not likely to have a major economic impact on the participant. This allows for a conclusion that gives credence to both sides of the “debate”:  yes, microfinance does have a significant positive effect on participants on average; but the effect is relatively small.  (I.e., a useful tool, but not a panacea – which Section 4.4 lays out even more explicitly.)  This point could usefully be made more strongly in the conclusions and abstract.

Line 355 and Intro:  Although for Mohammad Yunus and some early proponents of microfinance, poverty reduction was a main motivation, this proved overambitious (as this paper so well demonstrates), and over time “financial inclusion” and sustainability have become significant objectives for microfinance. That is, those just above the poverty line who are “unbanked” can benefit from access to microfinance (and other non-bank sources of finance – including mobile money and other digital financial services, these days). It is too narrow to place microfinance solely in the domain of poverty reduction, and the paper would benefit from recognizing the broader role of microfinance in financial access and inclusion. Indeed, micro and small enterprise finance remains a problem area, because such enterprises are often not well served either by banks or by MFIs.  So if micro/small entrepreneurs just above the poverty line have benefited from this program, why treat that as a negative – rather than another way in which microfinance can contribute to income and economic growth.

Line 357:  There is a fundamental misconception here that microfinance does or even can charge lower rates than commercial banks. Except in some cases of highly subsidized microfinance programs (especially in the early days), MFIs have always had to charge higher interest rates due to significantly higher costs.  Even the most efficient, largest MFIs have operating costs on the order of 15-20% of the loan portfolio, so have to charge higher rates to cover loan losses, inflation, and make a surplus to be sustainable [the focus on low interest rates for the poor neglects the whole sustainability side of the outreach-sustainability trade-off issue]. MFIs have been able to give the impression that their interest rates are lower than they really are by charging flat rates rather than on declining balance (so the annual percentage rate can be as much as double the apparent nominal rate). And poor people who have good investment opportunities (especially with quick turnover, such as trading) but lack capital are willing and able to pay these higher rates.

Line 366:  There doesn’t seem any reason to think that the benefits of microfinance loans have spread to the non-borrowers (except perhaps by boosting the overall economy in the area – but you have said that the benefits are too small to have much economic effect). Rather, it would be more appropriate here to recognize that there may have been some causality in the other direction – i.e., that non-poor better-off people are better able to participate in MF.

EDITORIAL

Line 154: “time suffered influence” is an awkward and unclear phrase.

Line 316:  “…null hypothesis of no significant difference.”

Line 317:  “reject the” instead of “deny”

This article requires copy-editing to improve the flow in English and correct grammatical errors. There are too many for me to attempt to specify them all. Subject-verb agreement is a persistent problem. Also confusion between “its” and “it’s”

Author Response

 Dear reviewer thank you very much for your detailed valuable suggestions and recommendation; I/we have made adjustments accordingly. Please find answers to your questions as follows.

Question:  A brief literature review that recognizes other relevant empirical research.

Answer:  A detailed and extensive review of literature is added according to your recommendations, please see edited version

ABSTRACT

Question:  It is not correct that “no recent study assesses microfinance effects on rural female agricultural entrepreneurs.” A quick Google search finds, among others:

“Rural women’s access to financial services: Credit, savings and insurance,” Diana Fletschner and Lisa Kenney, FAO, ESA Working Paper No. 11-07, 2011 “The effect of microfinance on the empowerment of women and its societal consequences: A study of women self-help-group members in Andhra Pradesh”, KNUT-ERLAND BERGLUND, Department of Economic History, Uppsala University,  Minor Field Studies-reports 2007:1 Effects of Microfinance on Agricultural Occupation: Case Study in Battambang Province, Svay Sopheana et al., DOI: 10.7763/IPEDR. 2012. V46. 13

Answer:  It was not correct to say, that no recent study evaluates microfinance effects on rural female agricultural entrepreneurs. More light has been shed on this issue in the literature review part of the study and many relevant and existing studies are cited.

Question:  Also, since less than 12% of the borrowers were in agricultural, wouldn’t it be better just to say “rural female entrepreneurs” and drop the “agricultural”.

 Answer:  We totally changed title and removed agricultural as very correctly only 12% belonged directly to agriculture. Please see edited version

Question:  Although the substantial bibliography lists a number of publications on microfinance and on women’s empowerment, these are only referenced in the Introduction – which does an adequate job of summarizing arguments that microfinance can help address poverty and empowerment and of noting that other studies have questioned that premise or found negative effects. There needs to be a Literature Review section that reviews more specifically what studies of microfinance and women entrepreneurs (in agriculture or MSMEs) have investigated and concluded, in order to set up what the authors would expect to find in the context being studied – or to state explicitly that they are going to investigate whether the positive or negative effect hypothesis holds in Pakistan (for the sample group).

Answer:   It was not appropriate not to include literature review part in the study, this has now been corrected and an extensive and detailed overview of many relevant studies has been presented that evaluates both negative and positive aspects of microfinance program on borrowers. Please see edited version

Question:  Use of data at a point in time to establish a control group raises a fundamental problem of causality. It is noted that the “same household cannot be analyzed simultaneously for both being participant and non-participant”  -- i.e., there is no before-and-after data for the treatment group. So non-participants are used to serve as a control group who did not receive the “treatment.” It is assumed that microfinance is the “treatment.” But one could just as easily view higher income as the “treatment” – in which case we are testing the reasonable hypothesis that higher income yields better access to microfinance. [However, one could also argue that, at some point, people with still higher incomes don’t need microfinance, because they can access banks. 

The authors need to be careful to admit that what is being observed is an association, and that causality could be in the other direction. But based on the literature and a conceptual framework, it is presumed that if higher income/consumption is observed, it would be an effect rather than a determinant of participating in microfinance. And, further, the difference-in-difference methodology helps to sort out the direction of causality.

 Answer:  You pointed out very correctly and that’s how we solved this problem in our study, we believe that higher income and consumption is in fact an effect of inclusion in microfinance program rather than determinant of participants in microfinance programs and direction of causality is sorted out to an acceptable and reasonable level through use of difference in difference methodology. Please see edited version

Question:  Some studies handle this problem by taking as the control group people who have joined the microfinance organization but have not yet received loans – thus they can be presumed to be comparable in other characteristics – including level of income/consumption at the time of entry to the organization -- to other members who have previously received loans.

This study uses the next-best approach, which is to observe both partipants and non-participants at two points in time, and to attribute any differences in their relative changes over time to the treatment variable. This is adequately explained on lines 158-162; and the flaw in the approach is recognized in lines 180-4, followed by a reasonable effort to deal with it.

However, it appears that the study does not actually gather data at two points in time. Rather, the ‘treatment group’ consists of participants who had a loan at the time of the survey but were not participants in the “pre-program period” (line 144).  Lines 240-2 clarify that 2009 was chosen to represent the “pre-program period” (to minimize any anticipatory effects in the lead-up to the 2011 launch) and 2016 for post-program. But doesn’t that mean that people were being asked to recall their consumption and incomes 8 years prior to the 2017 survey?!? (since there was no baseline survey). How reliable can recall data be for that previous period (especially for income, which is notoriously difficult to capture adequately through self-reported survey responses)? We need more detail on how income and consumption were measured, and the uncertainty involved if they were indeed based on recall data.

Answer:  It is very difficult for low income individuals (both men and women) to have proper book keep accounting records. Our respondents were small female entrepreneurs with poor literacy. It is a norm in the country not to keep any financial records, so small business owners (grocery shops, tailoring centers, handicrafts makers, beauty parlors, tuition centers, livestock and poultry farmers, small agricultural producers etc.) do not or cannot keep business records due to many reasons; illiteracy, lack of knowledge and skills, lack of time or shortage of labor and fear of corrupt tax/revenue officials. However, many have some sort of rough records in form of single entries in their private books, with sums like, total borrowings, total loan amount and interest, unpaid and paid loans, total earning and total home spending and savings.  It is against the law of the country to have an unregistered firm. In many instances even small limited companies lack proper book keep and financial records. Majority of microfinance studies done on low income borrowers in Bangladesh, India and other low income countries that are similar in characteristics to Pakistan are based on recall data. Unfortunately, this is the only way to record income and consumption of low income individuals based on survey responses, with the exception of macro studies done on impacts of microfinance at national and international scales that can make good use of public records recording microfinance institutions. But these studies in general are more provider focused than recipients and fails to capture issues and problems that borrowers face at micro level. Please see edited version

Question:  Also, assuming these were monetary values, how was inflation handled?

Answer:   Please see introduction section, efforts were being made to record values according to national adjusted inflation values. Please see edited version

Question:  Also, we should know the basis on which people were accepted into the program. As noted above, it may be that those who qualified for the program already had a higher level of income, education, skills, etc. (which can only partially be controlled for through the regression technique for the adjusted DD approach.

Answer:  Please see methodology part, more light has been shed on this aspect, unfortunately difference and difference methods do have many shortcomings but it is still considered adequately reasonable to do social sciences research. This problem has been tried to fix through adjusted DD method.

Question:  Line 89:  Is it an established fact that microfinance per se caused the suicides? It may be more prudent to say “alleged” if there was simply an association between suicide and being a microfinance client (among other factors). [In any case, it was the inability to repay loans (due to various factors) that is involved, not the availability of loans, savings etc. as such.

Answer:  It is definitely not correct to say, ‘it is an established fact that microfinance caused suicides’. It is corrected now. Please see edited version. Nonnative writers sometimes unintentionally, makes terrible mistakes :)

Question:  Lines 309-310:  Using loan size as a continuous variable for usage of microfinance instead of a dummy variable for participation is a useful innovation, with good results.

 Answer:  I’m glad you like loan size as continuous variable for usage of microfinance instead of dummy variable for participation; I was little bit skeptical of its use but after your positive comment I’m happy that I did so.

Question:  Lines 333-4 usefully clarify that, even though a statistically significant difference may be found, the magnitude of the difference is relatively small – and thus not likely to have a major economic impact on the participant. This allows for a conclusion that gives credence to both sides of the “debate”:  yes, microfinance does have a significant positive effect on participants on average; but the effect is relatively small.  (I.e., a useful tool, but not a panacea – which Section 4.4 lays out even more explicitly.)  This point could usefully be made more strongly in the conclusions and abstract.

Answer:  Please see revised abstract, literature review and conclusion section; many changes has been made to address this question. This has been explained in detail, in abstract and conclusion parts of the study.

Question:  Line 355 and Intro:  Although for Mohammad Yunus and some early proponents of microfinance, poverty reduction was a main motivation, this proved overambitious (as this paper so well demonstrates), and over time “financial inclusion” and sustainability have become significant objectives for microfinance. That is, those just above the poverty line who are “unbanked” can benefit from access to microfinance (and other non-bank sources of finance – including mobile money and other digital financial services, these days). It is too narrow to place microfinance solely in the domain of poverty reduction, and the paper would benefit from recognizing the broader role of microfinance in financial access and inclusion. Indeed, micro and small enterprise finance remains a problem area, because such enterprises are often not well served either by banks or by MFIs.  So if micro/small entrepreneurs just above the poverty line have benefited from this program, why treat that as a negative – rather than another way in which microfinance can contribute to income and economic growth.

 Answer:  This was in fact a mistake to treat above poverty level individuals who get benefited from use of microfinance services as negative aspect of microfinance programs, though they belong to above poverty level group of people but are still disadvantaged and excluded section of the society from mainstream financial institutions because of lack of collateral, skills and knowledge or initial capital requirements. Please see edited version

Question:  Line 357:  There is a fundamental misconception here that microfinance does or even can charge lower rates than commercial banks. Except in some cases of highly subsidized microfinance programs (especially in the early days), MFIs have always had to charge higher interest rates due to significantly higher costs.  Even the most efficient, largest MFIs have operating costs on the order of 15-20% of the loan portfolio, so have to charge higher rates to cover loan losses, inflation, and make a surplus to be sustainable [the focus on low interest rates for the poor neglects the whole sustainability side of the outreach-sustainability trade-off issue]. MFIs have been able to give the impression that their interest rates are lower than they really are by charging flat rates rather than on declining balance (so the annual percentage rate can be as much as double the apparent nominal rate). And poor people who have good investment opportunities (especially with quick turnover, such as trading) but lack capital are willing and able to pay these higher rates.

 Answer:  Indeed it would not be a viable and sustainable model for microfinance to charge lower interest rates without any government subsidies and grants. Therefore, it has now been corrected and adjustment are made in abstract and conclusion parts of the study according to your recommendations. Its been stressed that masses below poverty line or poorest of the poor should be addressed according to their financial profiles and should be helped out through other means (grants, health insurance, basic skill training and food stamps etc.) Please see edited version

Line 366:  There doesn’t seem any reason to think that the benefits of microfinance loans have spread to the non-borrowers (except perhaps by boosting the overall economy in the area – but you have said that the benefits are too small to have much economic effect). Rather, it would be more appropriate here to recognize that there may have been some causality in the other direction – i.e., that non-poor better-off people are better able to participate in MF.

 Answer:  Again this was not correct to say so, and rightly now this has been corrected and is explained in conclusion and abstract that non poor who’re still excluded from mainstream financial institutions due to their low financial profile actually benefit more from microfinance programs. Please see edited version

EDITORIAL

Question:  Line 154: “time suffered influence” is an awkward and unclear phrase.

Answer:   Its been corrected. Please see edited version

Question:  Line 316:  “…null hypothesis of no significant difference.”

Answer:  It was grammatical mistake, corrected now. Please see edited version

Question:  Line 317:  “reject the” instead of “deny”

Answer:  Corrected accordingly. Please see edited version

This article requires copy-editing to improve the flow in English and correct grammatical errors. There are too many for me to attempt to specify them all. Subject-verb agreement is a persistent problem. Also confusion between “its” and “it’s”

Answer:  Its always good to read and reread for spelling and grammatical mistakes. Now I’ve re edited the whole manuscript and checked it thoroughly for mistakes. Please let me know if you think it still lacks adequate punctuation and flow, in that case, I’ll edit it through a native English speaker.

Note:  I’m stuck in Wuhan China (ground zero of corona virus). The work on this paper did help me to distract, my attention from the ongoing situation of shutting down of Hubei province; and the many negative consequences it has caused to routine activities of people in the this area.

Reviewer 2 Report

The topic is interesting and could offer a great contribution to the scientific world if it was better designed. Yet, the title is 'Microfinance Facility For Women Enterpreneurs In Pakistan: Friend or Foe?' and the analysis is based on borrowers - non-borrowers differences. I strongly advice to include in the research methodology a differentiation between men and women borrowers, otherwise the purpose of your research is not completed. 

Also, a table explaining the indicators that you calculated is necessary for the readers, so they can understand the meaning of those numbers. 

Microfinance is a term that you use in your research, but there is no definition of it in the paper. You should also include that. 

The software that you used for calculation should be mentioned, so other researchers can replicate your research for other sets of data. 

You should include a correlation of your results with other existing scientific results.

The literature review should consider more perspectives on this topic. 

Author Response

Dear reviewer thank you for your valuable suggestions and recommendation; I/we have made adjustments accordingly. Please find answers to your questions in as follows.

Question:  Yet, the title is 'Microfinance Facility For Women Enterpreneurs In Pakistan: Friend or Foe?

Answer:  Title has been adjusted according to your observations. Please see edited version.

Question:  I strongly advice to include in the research methodology a differentiation between men and women borrowers, otherwise the purpose of your research is not completed. 

Answer:  Please see revised methodology section for regarding the issue of differentiation in men and women borrowers and the importance and significance of women borrowers in added review of literature section.

Question:  Also, a table explaining the indicators that you calculated is necessary for the readers, so they can understand the meaning of those numbers. 

 Answer. Please see an explanation of the main terms used in table 1, below table 1. Please see edited version  

Question:  Microfinance is a term that you use in your research, but there is no definition of it in the paper. You should also include that. 

Answer:  It was a mistake not to include definition of microfinance. Definition of microfinance is now included in the revised article. Please see edited version

Question:  The software that you used for calculation should be mentioned, so other researchers can replicate your research for other sets of data. 

Answer:  I/we forgot to add details of software used for analytical analysis. This mistake is now corrected. Please see edited version

Question:  You should include a correlation of your results with other existing scientific results.

Answer:  Please see added review of literature section; many relevant studies with similar findings are 

               quoted.

Question:  The literature review should consider more perspectives on this topic. 

Answer:  Please see, new added section ‘review of literature’, an extensive and detail analysis of microfinance is presented.

Note:  I’m stuck in Wuhan China (ground zero of corona virus). The work on this paper did help me to distract, my attention from the ongoing situation of shutting down of Hubei province; and the many negative consequences it has caused to routine activities of people in the this area.

Reviewer 3 Report

The theme of the paper is interesting, and its a current topic. The paper is well written and structured. 

Please pay attention to formatting requirements. You use in some parts  "by 0.26%..." why the dots? Are they neccesary? Under table 6, line 352 you write "x the low poverty line is less than $1 for both 2009 and 2016." and "the low income level is less than $1.50 for both 2009 and 2016." line 353. This should be a note for the table? If so, please mention this. Please describe which is the main purpose of the paper and its added value to the literature;  at the end of the introduction section you should present the structure of the paper;  the limitations of the present study and future directions of improvement should be presented in conclusion section; present a graph or a table that summarizes the main variables used in the empirical analysis;

Author Response

 Dear reviewer thank you for your valuable suggestions and recommendation; I/we have made adjustments accordingly. Please find answers to your questions in as follows.

Question:  You use in some parts  "by 0.26%..." why the dots? Are they neccesary

Answer:  Unfortunately, I did not recheck article for spellings and grammatical mistakes. The dots were grammatical mistake, corrected and removed now. (please see edited version)

Question: Under table 6, line 352 you write "x the low poverty line is less than $1 for both 2009 and 2016." and "the low income level is less than $1.50 for both 2009 and 2016." line 353. This should be a note for the table?

Answer:  Again, forgot to put a note below the able. Please see the edited version of the study, its been corrected now.

Question:  Please describe which is the main purpose of the paper and its added value to the literature?

Answer:  Main purpose of the study is emphasized more in abstract, methodology and conclusion sections in detail. (please see edited version)

Question: Present a graph or a table that summarizes the main variables used in the empirical analysis?

Please see conceptual framework for graphical representation of the study. Also the main variables used are explained below table 1.

Note:  I’m stuck in Wuhan China (ground zero of corona virus). The work on this paper did help me to distract, my attention from the ongoing situation of shutting down of Hubei province; and the many negative consequences it has caused to routine activities of people in the this area.

Round 2

Reviewer 2 Report

The additions to the material are very important. I can see the improvement.